# Non-Reproducibility of Oral Rotenone as a Model for Parkinson’s Disease in Mice

**DOI:** 10.3390/ijms232012658

**Published:** 2022-10-21

**Authors:** Ellen Niederberger, Annett Wilken-Schmitz, Christine Manderscheid, Yannick Schreiber, Robert Gurke, Irmgard Tegeder

**Affiliations:** 1Institute for Clinical Pharmacology, Goethe-University Frankfurt, Theodor Stern-Kai 7, 60590 Frankfurt, Germany; 2Fraunhofer Institute for Translational Medicine and Pharmacology ITMP, Theodor Stern-Kai 7, 60596 Frankfurt, Germany; 3Fraunhofer Cluster of Excellence for Immune Mediated Diseases CIMD, Theodor Stern-Kai 7, 60596 Frankfurt, Germany

**Keywords:** Parkinson’s disease, rotenone, oral, mice, motor behavior, synuclein

## Abstract

Oral rotenone has been proposed as a model for Parkinson’s disease (PD) in mice. To establish the model in our lab and study complex behavior we followed a published treatment regimen. C57BL/6 mice received 30 mg/kg body weight of rotenone once daily via oral administration for 4 and 8 weeks. Motor functions were assessed by RotaRod running. Immunofluorescence studies were used to analyze the morphology of dopaminergic neurons, the expression of alpha-Synuclein (α-Syn), and inflammatory gliosis or infiltration in the substantia nigra. Rotenone-treated mice did not gain body weight during treatment compared with about 4 g in vehicle-treated mice, which was however the only robust manifestation of drug treatment and suggested local gut damage. Rotenone-treated mice had no deficits in motor behavior, no loss or sign of degeneration of dopaminergic neurons, no α-Syn accumulation, and only mild microgliosis, the latter likely an indirect remote effect of rotenone-evoked gut dysbiosis. Searching for explanations for the model failure, we analyzed rotenone plasma concentrations via LC-MS/MS 2 h after administration of the last dose to assess bioavailability. Rotenone was not detectable in plasma at a lower limit of quantification of 2 ng/mL (5 nM), showing that oral rotenone had insufficient bioavailability to achieve sustained systemic drug levels in mice. Hence, oral rotenone caused local gastrointestinal toxicity evident as lack of weight gain but failed to evoke behavioral or biological correlates of PD within 8 weeks.

## 1. Introduction

Parkinson’s disease (PD) is the second-most common neurodegenerative disease worldwide after Alzheimer’s disease, with increasing prevalence with advancing age [1]. Symptoms of the disease are mainly tremor, movement disorders, and rigidity. Less-common symptoms include motor disorders such as dystonia, dysphagia, and non-motor problems such as dementia, anxiety, depression, and pain [1,2,3]. A hallmark feature of PD is a loss of dopaminergic neurons, particularly in the substantia nigra but also in other brain areas, and the accumulation of Lewy bodies in the surviving neurons. These consist mainly of aggregated alpha-synuclein (α-Syn, Snca). In Lewy bodies, aggregated α-Syn is present in a fibrillar form whose formation is promoted by oxidative stress or mitochondrial malfunctions and defective post-translational modifications of the native protein [4,5]. In the brains of Parkinson’s disease patients and of rodents in Parkinson’s disease models, dysregulation of α-Syn, including oligomerization and fibril formations, are common manifestations contributed by lysosomal defects of aggregate removal [6,7,8]. It has been suggested that α-Syn proteins may spread from peripheral sensory and autonomic neurons and behave like prions [9,10]. To date, there is no cure for the disease and its molecular mechanisms are still incompletely understood, in part owing to limitations of available rodent models. Several animal models of PD have been developed in rats and mice. They are mostly genetic models of human mutant synuclein or knockouts of key Parkinson’s-associated genes such as Pink1 or Parkin [11,12,13,14]. Alternatively, local or systemic MPTP (1-methyl-4-phenyl-1,2,3,6-tetrahydropyridin) or other dopaminergic neurotoxins have been widely used [15,16]. All of these models replicate parts of human PD pathology, but the behavioral manifestations are mostly subtle and the predictive nature for human PD is still limited.

Rotenone is a natural ingredient of Leguminosa plants that inhibits complex I of the mitochondrial respiratory chain. It is commercially used as a pesticide and able to cross the blood–brain barrier due to its high lipophilicity. The rotenone PD model was originally developed in rats which showed PD-like phenomena associated with dopaminergic neuron degradation and formation of Lewy body-like structures in the brain after intravenous administration of the substance [17]. A problem with the model is its high mortality, which hampers reproducibility of the results and offends 3R criteria. To reduce animal suffering and mortality, alternative methods of administration including oral, subcutaneous, and intraperitoneal have been investigated and were described as being reproducible and reliable [18,19,20,21,22]. Oral administration has also been established in mice [21,22] and was used as a model in multiple experimental studies investigating novel therapeutic compounds against PD [23,24,25,26,27,28,29,30]. Therefore, we adopted the described high-dose treatment regimen of oral rotenone in mice exactly following the published protocol, with RotaRod running as the behavioral and tyrosine hydroxylase immunostaining as the biological readouts [21,22]. Our aim was to study long-term behavioral and morphologic outcomes. However, plasma pharmacokinetic studies revealed that the drug was not sufficiently bioavailable, and accordingly mice showed no PD-like phenomena but only lack of weight gain, likely resulting from high local intestinal drug exposure and toxicity on the gut and microbiome [31,32,33,34,35].

## 2. Results

### 2.1. Weight Gain and Health in Rotenone and Vehicle Group

Mice were treated with rotenone (30 mg/kg body weight, p.o.) and vehicle 5 times per week. The health score, including general appearance and eating and drinking behavior, as well as movement and weight gain, were monitored daily with the exception of weekends. Rotenone and vehicle-treated mice did not show any impairments of well-being throughout the observation time. All mice were healthy and there were no drop-outs, which is in contrast to other studies showing more or less high mortality (up to 50%) under rotenone treatment in mice. Mice were 6–8 weeks old at the onset of treatments, and as expected vehicle-treated mice constantly gained weight during the observation period. In contrast, rotenone-treated mice remained at their starting body weight. Time courses differed significantly between groups, but one-way ANOVA of final body weights did not reach statistical significance, owing to high variability in the rotenone group (Figure 1) (Table 1).

### 2.2. Motor Function

Motor function was assessed two times per week during the treatment period using the RotaRod test. The running times (fall-off latencies) did not differ between groups at any time point, indicating that the rotenone treatment did not evoke impairments of motor coordination, endurance, or running motivation. Figure 2 shows the individual running times for both groups during the complete observation period (mean of two runs per test day) as well as the running on the last day of the treatment period of 28 days or 54 days. Appendix A further depicts the individual time courses of each single run sequentially, and the distribution of all pooled runs in comparison with a former control group of mice, which had similar ages at observation start, also did replicate runs per test day, and were observed over a similar period. The previous age-matched control mice show a similar running behavior as vehicle and rotenone-treated mice.

### 2.3. Immunofluorescence

In addition to behavioral analysis, we assessed the effects of rotenone on dopaminergic neurons and potential accumulation of α-Synuclein using immunofluorescence studies of brain sections. Furthermore, potential changes in astrocytes and microglia in the brain were also investigated using immunofluorescence analysis. There was no accumulation of α-Syn in the substantia nigra or other brain regions in rotenone-treated mice. There was also no loss of dopaminergic neurons in the substantia nigra (as revealed by immunofluorescence analysis of tyrosine hydroxylase) or immune cell infiltration, in agreement with previous reports [21]. Morphologic features of microglia in the substantia nigra were alike in both groups and mostly agreed with resting state morphology, but quantitative analysis of CD11b immunoreactivity revealed higher numbers of CD11b-positive cells in rotenone-treated mice, suggesting mild microgliosis (Appendix A). Considering the absence of α-Syn and absence of TH differences in the SN (Figure 3, Appendix A), mild microgliosis would agree with a subtle gut-to-brain effect in response to rotenone-mediated disruption of gut homeostasis [36,37].

### 2.4. Rotenone Plasma Concentrations

Since we did not observe differences in the treatment groups concerning health, RotaRod performance, and histology, we assessed plasma concentrations 2 h after the last rotenone dose to reveal a putative pharmacokinetic (PK) failure. Low bioavailability of oral rotenone has been described before [15]. Rotenone concentrations in the plasma of mice was determined by liquid chromatography combined with tandem mass spectrometry. The lower limit of quantification (LLOQ) was 2 ng/mL (5 nM). Concentrations of rotenone were below detection limit (10× lower than LLOQ) for all samples except one, which showed a measurable concentration of 0.964 ng/mL, but still below the LLOQ. The PK results show low bioavailability of oral rotenone irrespective of its high lipophilicity, either owing to low absorption (unlikely) or very fast and near complete first-pass metabolism. Drug levels in plasma were too low for any systemic effect. Considering the large volume of distribution it cannot be excluded that small amounts might have accumulated in tissue, including the brain, but our “no-weight-gain” data suggest that oral rotenone underwent fast and near complete first-pass metabolism in the gut and liver, where it caused GI toxicity as described [31,32,34], preventing mice from gaining body weight; it is very unlikely that orally administered rotenone reached the central nervous system.

## 3. Discussion

The aim of the study was to establish a temporally well-controlled, dose-dependent, reliable, and reproducible model for Parkinson’s disease in mice that would be compatible with 3R criteria and would replicate at least in part PD-typical pathology and the motor-function deficits of human PD. Systemic administration of rotenone has been described in several publications as a promising, relatively novel approach to phenocopying the slowly progressive course of the disease in rats and mice. In particular, studies used i.p., s.c., and p.o. as well as intranasal and dermal exposure to the drug, with a number of different readouts to assess histologic, biochemical, and in vivo correlates of human Parkinson’s disease [21,22,38,39,40,41,42]. Comparisons of the results of these studies reveal high within- and between-study variability even with very similar protocols, but particularly high-dose oral rotenone looked promising and has been used as a PD model in multiple studies exploring drug, diet, stress, or LRRK2 effects in mice [23,25,26,27,28,29,30,36,39,43,44,45]. Interestingly, studies addressing drug or diet effects found quite stable reductions in RotaRod running of about 50% and restoration with the candidate drug [23,25,26,27,28,29,30], whereas studies addressing add-on effects of stress or LRRK2 mutation found minor or no effect of rotenone alone but a serious drop in combination [36,39,45], suggesting aim-dependent biases. Mortality rates are not always reported, but high numbers of dropouts may preclude the most severe cases from final analysis. In addition, pharmacokinetic features likely depend on the route of drug administration, leading to strong variations in rotenone concentrations in plasma or brain. Rotenone’s PK parameters have not been systematically compared in rodents, and most of the rotenone PD studies did not analyze plasma levels of rotenone. In one study where plasma concentrations were determined, levels were undetectable in some cases at the onset of in vivo symptoms [46]. A further study found no rotenone in the brain at 10 mg/kg/d p.o [36]. It is not clear how PD-like pathology may arise in the absence of measurable plasma and brain concentrations. It has been suggested that oral rotenone disrupts the gut microbiome and intestinal barrier [30,32,33,34,35] and leads to accumulation of α-Syn in the enteric nervous system (ENS) [31,43] from where it is supposed to spread to the brain via the vagus, which has been experimentally demonstrated by direct injection of α-Syn into the vagus nerve [47,48,49]. Our “no-weight-gain” data agree with rotenone-triggered gut pathology and suggest that rotenone undergoes first-pass metabolism in the gut and liver, in agreement with case reports of rotenone fatalities in humans [50] and liver toxicity in rats [51]. The toxicity of the drug in the gastrointestinal tract and liver likely causes sickness behavior in mice that might manifest in reduced RotaRod running in some studies. Gut dysbiosis, barrier leakage, and local mitochondrial damage may promote α-Syn accumulation in the ENS and may cause gut-to-brain proinflammatory signaling, resulting in microgliosis. According to the hypothesis of Braak [52,53], rotenone-evoked PD-like phenomena are a result of gastrointestinal accumulation and a retrograde transmission of α-Syn via the ENS to the brain, which would agree with α-Syn prion-like spreading [54], and lack of weight gain in our rotenone-treated mice agrees with the autonomic non-motor symptoms of PD. Beyond these effects on body weight, there was no evidence of drug-evoked toxicity in our mice except mild quantitative microgliosis. None of the mice had health problems or died, which is in accordance with previous toxicology studies [55] but does not agree with previous rotenone-PD studies in which up to 50% of animals dropped out [21,56]. Our data further indicate that orally administered rotenone fails to manifest in measurable plasma concentrations, likely owing to fast first-pass metabolism. Considering its lipophilicity, non-absorption is less likely. In our mice, toxic effects on the gastrointestinal tract were however not associated with α-Syn accumulation or toxicity on dopaminergic neurons in the brain, either because the effect was too weak or the time frame insufficient to allow for spreading of α-Syn to the brain. Consequently, there was also no effect of rotenone on the motor behavior in the RotaRod test. This lack of effect after oral rotenone has also been described in another study, which focused mainly on the anticarcinogenic effects of rotenone in a rat model. Similar to our results, the authors described retarded weight gain in rotenone-treated mice but no evidence for rotenone-induced neurotoxicity after oral administration (52 mg/kg body weight for 14 d) and the authors pointed to low oral bioavailability as the most likely explanation [57]. Further studies found that rotenone-evoked manifestations of neuropathology require add-on stressors such as restraint stress, aging, or LRRK2 mutations [36,39]. It is a weakness of our study that sample sizes were low, particularly for the treatment period from day 28 to day 54. Based on the previous protocol, the study was powered for an observation time of 28 d and was limited by ethical restrictions resulting from assumptions of high mortality. To assess biological effects at 28 d (previous endpoint) half of the animals had to be euthanized at 28 d, leaving only small groups for observation up to 54 d. Hence, statistical comparisons of behavioral data of the final period are hampered by low sample sizes. Nevertheless, comparison of individual behavioral RotaRod data suggest a mild advantage of rotenone-treated mice versus vehicle-treated mice, possibly owing to the lower body weight and no difference in comparison with a former control group. Irrespective of the low sample size, our pharmacokinetic studies clearly reveal that rotenone is not bioavailable via the oral route, so the CNS is not directly exposed.

Nevertheless, the majority of oral rotenone-PD studies claimed the model as suitable [21,23,25,26,27,28,29,30,33,34,35,37,43,44,58,59]. Some of these studies used different solvents, such as sunflower oil 4% or 2% carboxymethylcellulose with 1.25% chloroform, which likely affects absorption [59,60,61,62] but not so much first-pass metabolism. According to SwissADME prediction (http://www.swissadme.ch, accessed on 6 September 2022), GI absorption is per se high, and rotenone is substrate and inhibitor of cytochrome P450 enzymes. Therefore, PK is difficult to predict, particularly with once-daily treatment. Nevertheless, other reports used exactly the same treatment protocol as chosen in our study. It is not clear why oral rotenone produces dopaminergic neuron degeneration, α-Syn accumulation, and motor deficits in some studies but not in others. Details of methodology, sample sizes, and methods of randomization and blinding may contribute to high inter-study variability and low reproducibility.

These problems and differences further emphasize the need for more standardized protocols in which the vehicles, the sources of drugs, composition of drug and vehicle solutions/suspensions and dosing schedules, and genetic background of the animals are described as precisely as possible to increase comparability and reproducibility of the studies. In addition, rotenone-standardized diets (food pellets) may overcome the problem of fluctuating concentrations and rapid pre-systemic metabolism, and oral rotenone may indeed be useful to study PD-associated autonomic neuropathy of the ENS.

## 4. Materials and Methods

### 4.1. Animals

Male C57BL/6J mice were obtained from Charles River, Sulzfeld, Germany at the age of 6–8 weeks. Animals had free access to food and water and were maintained in climate- and light-controlled rooms (24 ± 0.5 °C, 12/12 h dark/light cycle). All behavioral experiments were performed by an observer blinded for the treatment in a dedicated room with restriction on sound level and activity.

Ethics Statement: Animal experiments adhered to the ethical guidelines for investigations in conscious animals, and the procedures were approved by the local Ethics Committee for Animal Research (Regierungspräsidium Darmstadt, Germany, permit no. FK1136). All efforts were made to minimize animal suffering and reduce the number of animals according to 3R principles.

### 4.2. Reagents

Rotenone was purchased from Sigma-Aldrich (Darmstadt, Germany). For animal experiments, it was suspended in 0.5% carboxymethyl cellulose sodium salt (Sigma-Aldrich, Darmstadt, Germany) with 0.1% Tween-20 (Carl Roth, Karlsruhe, Germany). The suspension was freshly prepared every day. The rotenone and the 0.5% carboxymethyl cellulose sodium salt/0.1% Tween-20 solution were stored in the dark at room temperature.

### 4.3. Animal Treatment

Mice were treated orally with vehicle or rotenone (30 mg/kg body weight) for 5 consecutive days a week and were drug-free on weekends. The treatment period was 28 days for 10 mice in the rotenone group and 6 mice in the vehicle group. At day 28, half of the mice were euthanized for histological analysis and the other half was continued up to 54 days. Based on previous studies, the primary endpoint was 28 d and the study was powered for this endpoint. Owing to the expectation of high mortality, only low sample sizes were ethically approved. Behavioral data of the vehicle group were supported/bolstered with former behavioral data of a control group, which was similar in age, RotaRod testing frequency, observation period, and identical C57BL/6 genetic background. The data were available from a previous study investigating motor behavior not related to PD. General health and weight gain were monitored daily before drug administration. Twice a week, the mice were subjected to a RotaRod test to investigate motor functions and coordination.

### 4.4. RotaRod Test

Motor coordination, endurance, and motivation were assessed with an accelerating-speed RotaRod for mice (Ugo Basile, Comerio, Italy) in an accelerating mode with speed increasing from 4 to 40 rpm over 5 min. All mice had four training sessions before the first day of the experiment. Two running series were performed every test day. 

### 4.5. Determination of Rotenone Plasma Concentrations

Animals were killed by CO_2_ and cardiac puncture 2 h after the last rotenone dose. For plasma preparation, blood was collected in EDTA tubes and centrifuged at 2000× *g* for 90 s. After centrifugation, the plasma was transferred to a fresh tube and stored at −80 °C until further analysis.

Rotenone was quantified in plasma samples using an LC-MS/MS method. All solvents were LC-MS grade. A gradient elution with 10 mM ammonium formate + 0.1% formic acid (A) and acetonitrile with 0.0025% formic acid (B) was run on an Agilent 1200 LC system (Agilent, Waldbronn, Germany) with a flow rate of 400 µL/min using a Zorbax C8 Eclipse Plus RRHD column (50 × 2.1 mm, 1.8 µm, Agilent, Waldbronn, Germany) with a precolumn for analyte separation in 7.5 min. MS/MS analysis was performed on a QTRAP 5500 triple quadrupole mass spectrometer (Sciex, Darmstadt, Germany) in positive ion mode, using the following transitions: *m*/*z* 395.1 > 213.0 (quantifier) and 395.1 > 192.0 (qualifier) for rotenone and *m*/*z* 247.2 > 204.1 (quantifier) and 247.2 > 202.0 (qualifier) for carbamazepine-D_10_ as the internal standard. MRM and MS parameters were optimized to achieve the highest signal yield.

The internal standard (20 µL, 5 ng/mL in MeOH) was added to 20 µL of thawed plasma samples, which were then purified using liquid–liquid extraction with ethyl acetate. The organic phase was dried under nitrogen at 45 °C, reconstituted with 50 µL methanol, and 10 µL were injected into the LC system. Calibration standards covering a range from 2.0 up to 200.0 ng/mL were created by adding appropriate working solutions to K3EDTA plasma not containing any rotenone. Absence of rotenone in the plasma for the calibration curve was evaluated by analyzing blank (no analyte or IS) and zero (no analyte) samples. Method verification included analysis of spiked plasma samples as calibration curve at levels from 2 ng/mL to 200 ng/mL in combination with measuring quality control samples at low, medium, and high levels. Quality control measures were performed during every sample run and included two sets with low-, medium-, and high-quality control samples. Acceptance criteria for accuracy was set to 20%. The lower limit of quantification (LLOQ) in 20 µL plasma, defined as signal-to-noise ratio of ≥10, was 2 ng/mL (S/N ratio determined for the LLOQ: 32.8). Limit of detection (LOD) was defined by a signal-to-noise ratio of ≥3. Quality control measures were performed during every sample run and included two sets with low-, medium-, and high-quality control levels. Data acquisition and evaluation was performed using Analyst 1.7.1 and MultiQuant-Software 3.0.3 (both Sciex, Darmstadt, Germany).

### 4.6. Immunofluorescence

Mice were euthanized with CO2, blood was collected by cardiac puncture, and mice were then cardially perfused with 1× PBS followed by 2% PFA in 1× PBS for fixation. The brains were collected for immunofluorescence staining. They were placed in 2% PFA in 1× PBS for 24 h for post-fixation, and then transferred to 20% sucrose solution for at least five hours and stored in 30% sucrose solution overnight at 4 °C for cryoprotection. The tissues were then embedded in cryomedium (Tissue-Tek O.C.T. Compound, Sakura Finetek Europe B.V., Alphen aan den Rijn, The Netherlands), frozen on dry ice, and cut into 16 μm thick cryosections (cross sections) at −21 °C. The slides were stored at −80 °C until histological staining.

For antibody staining, sections were washed in PBS and then incubated in PBSTx (1× PBS + 0.1% Triton X-100) for 10 min, followed by incubation in blocking solution (PBSTx + 3% BSA + 10% normal goat serum (NGS)) at room temperature for 60 min. Afterward, primary antibodies against tyrosine hydroxylase (Sigma, Deisenhofen, Germany, 1:200), alpha-Synuclein, (BD Bioscience, Franklin Lanes, Evansville, IN, USA, 1:100), CD11b (WAKO, Neuss, Germany, 1:100), and GFAP (Sigma, Deisenhofen, Germany, 1:1000) in PBSTx + 3% BSA were applied at 4 °C overnight in PBSTx. After washing with PBSTx for 3 × 10 min, Cy-3 and Alexa Fluor 488-conjugated secondary antibodies (Molecular Probes; Eugene, OR, USA; 1:1000) were applied for 2 h at RT. After a final washing step, the sections were fixed with mounting medium (Aqua Polymount, Polysciences Warrington, Bucks County, PA, USA) and cover slips. Microscopy was performed using an inverse fluorescence microscope (Zeiss Axioimager, Zeiss, Jena, Germany).

Images were analyzed in FIJI ImageJ. RGB images were converted to 8-bit images. Brightness and contrast were adjusted if necessary. The pseudo-flat-field correction plugin was used to adjust uneven illumination, followed by background subtraction. Images were converted to binary images using the IJ-IsoData threshold algorithm for tyrosine hydroxylase and CD11b and Yen’s algorithm for α-Syn and GFAP. Immunofluorescent particles were analyzed using the particle analyzer. Binary masks are presented as Appendix A. The percentage area was used for group-wise comparisons.

### 4.7. Data Analysis

Statistical evaluation was performed using GraphPad Prism 9 (GraphPad Software Inc., San Diego, CA 92108, USA). Data are presented as mean ± SD. Data were either compared by univariate analysis of variance (ANOVA) with subsequent t-tests employing a Dunnett’s correction for multiple comparisons versus vehicle-treated mice or baseline or by Student’s *t*-test. Non-parametric alternatives were used for small sample sizes (immunofluorescence). Time courses of body weights were submitted to ANOVA for repeated measurements (rmANOVA). For all tests, a multiplicity-adjusted probability value of *p* < 0.05 was considered statistically significant.

## Figures and Tables

**Figure 1 ijms-23-12658-f001:**
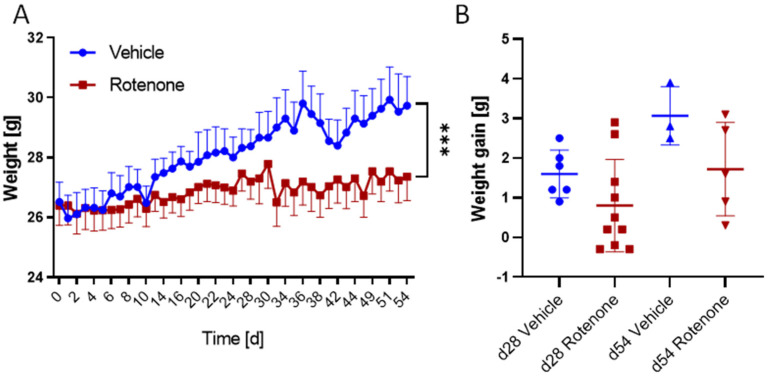
Weight gain in rotenone- versus vehicle-treated mice. (**A**) Time course of body weights from day 0 to day 54. (**B**) Body weight difference from treatment onset to day 28 or 54 (weight gain in grams) in vehicle and rotenone-treated mice, mean ± SD. Scatters show individual mice. Sample sizes: from day 0 until day 28: n = 6 for vehicle; n = 10 for rotenone, from day 28 until day 54: n = 3 vehicle; n = 5 rotenone. *** *p* < 0.001, statistically significant difference between the treatment groups (rm2-way ANOVA).

**Figure 2 ijms-23-12658-f002:**
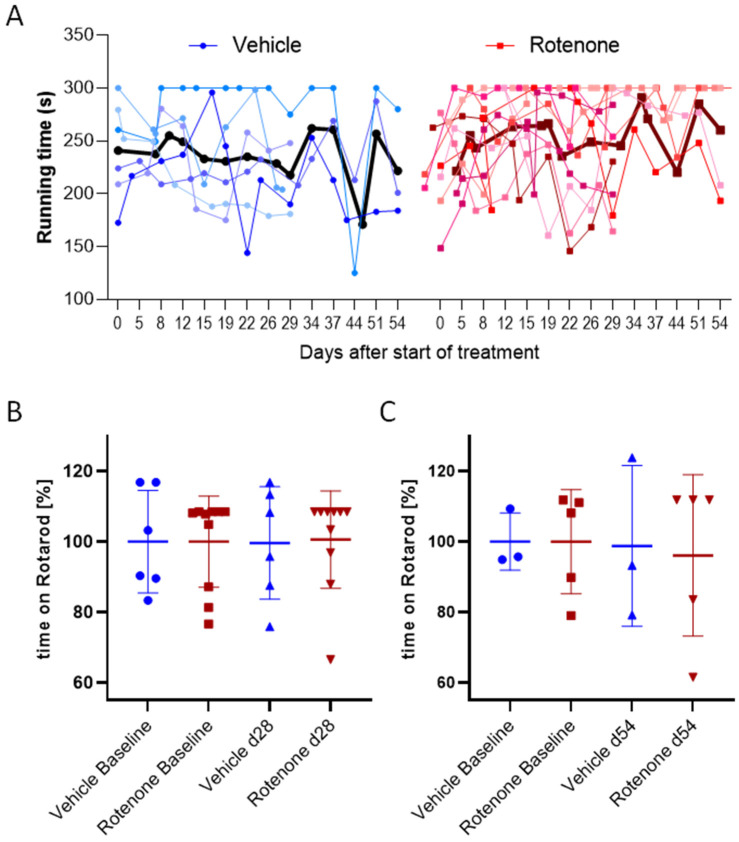
Running performance on an accelerating RotaRod, (**A**) Time course of RotaRod running times of individual vehicle- and rotenone-treated mice. Each thin line presents the mean of two runs/mouse/day, the thick lines represent the mean of all mice/group/day (red lines represent individual rotenone-treated mice, blue and purple lines vehicle-treated mice, the bold black line represents the mean of all vehicle mice). (**B**,**C**) Relative running times at (**B**) 28 d or (**C**) 54 d of rotenone or vehicle treatment as compared to mean baseline values of all mice set as 100%, mean ± SD. Each scatter is one mouse. Sample sizes: day 28: n = 6 vehicle, n = 10 rotenone, day 54: n = 3 vehicle; n = 5 rotenone.

**Figure 3 ijms-23-12658-f003:**
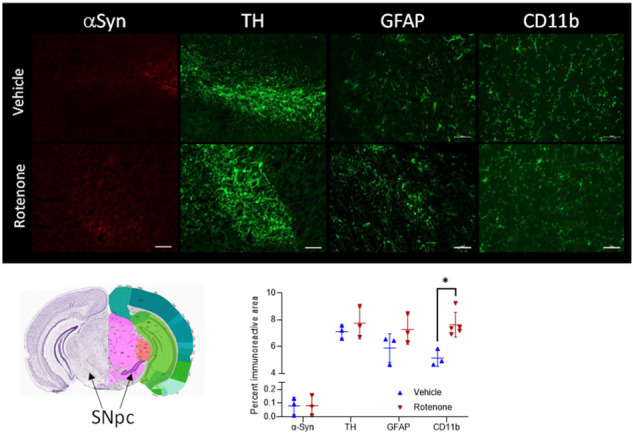
Immunofluorescence studies showing (from **left** to **right**) staining for alpha-Synuclein (α-Syn, red), tyrosine hydroxylase (TH, green) for dopaminergic neurons, glial fibrillary acid protein (GFAP, green) for astrocytes, and CD11b (green) for microglial cells and myeloid derived immune cells in the substantia nigra of vehicle- and rotenone-treated mice (54 d). The brain picture indicates the brain region which was investigated (substantia nigra pars compacta, SNpc), Allen Brain Atlas (https://mouse.brain-map.org/static/atlas, accessed on 3 August 2022). The diagram shows the quantitative analysis of the different proteins. Representative staining from at least n = 3 in the respective groups. Scale bar: 50 µm. * Statistically significant difference between vehicle and rotenone group, *p* < 0.05.

**Table 1 ijms-23-12658-t001:** Survival rates, weight gain and RotaRod performance of mice treated with vehicle or rotenone for 28 days or 54 days. BL = Baseline.

	Vehicle Day 28 (n = 6)	Rotenone Day 28 (n = 10)	Vehicle Day 53 (n = 3)	Rotenone Day 54 (n = 5)
Survival [%]	100	100	100	100
Weight gain [g] Mean ± SD	1.6 ± 0.6	0.8 ± 1.1	3.1 ± 0.6	1.7 ± 1.1
RotaRod [% BL] Mean ± SD	99.6 ± 14.6	100.6 ± 13.1	98.7 ± 20.5	96.1 ± 20.5

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
