# Peer review of "Non-Reproducibility of Oral Rotenone as a Model for Parkinson’s Disease in Mice"

_ijms, 2022, doi:10.3390/ijms232012658_

Round 1

Reviewer 1 Report

This is a nicely planned study to investigate the soundness of some of the existing animal models of PD. Reproducibility is a key question and this report casts doubt on some of the earlier papers demonstrating the effects of rotenone via the oral route. Such papers are needed for clarification of methodologies.

Minor point:

The authors should include a caveat of the very small numbers of animals used for the extended time points.

Author Response

We thank the reviewer for evaluation of our manuscript and the helpful feedback. We are aware of the low sample size for the late time points. The experiment was meant to establish the model, and the Institutional Review Board approved only small sample sizes and requested that the experiment exactly followed a published protocol which described strong effects of rotenone within 28 days. 
Therefore, we started the experiment with 6 animals in the control group and 10 animals in the rotenone group which were all observed up to 28 d. At d28, half of the animals were killed for histological analysis and the remaining animals were treated for further 26 days. According to the published protocol, rotenone effects should be evident within 28d in terms of survival, Rotarod running, increase of -Syn-positive neurons as well as a reduction of TH-positive neurons in the SN. Hence, to assess/confirm effects at this time point, we had to euthanize half of the animals at 28 days for histology and biological readouts. 
As requested by the reviewer we have added information to the Methods and the Discussion section in which we provide the justification for the low animal number in the later treatment period. 
In addition, we have now included in Suppl. Fig. 1 Rotarod running time courses of past control mice from a former experiment. These "ancient" controls were alike in terms of ages, genetic background and observation periods.

Reviewer 2 Report

Niederberger and colleagues describe a set of studies in mice that attempt to validate (or invalidate) the replicability of chronic oral rotenone model of Parkinson's in mice. The manuscript is well written and the methods section is detailed. The authors claim that because they were not able to show motor deficits using a rotenone dosing regimen used in other studies that the model is not reproducible, despite seeing gliosis and systemic toxicity as indicated by weight differences between groups. Understanding the limitations to reproducibility across labs is very important. I would also not be surprised if this rotenone is indeed not fully reproducible across labs. However, I do not believe the authors can make this claims with an underpowered study. Namely, at the later d54 timepoint, there were only 3-5 mice per group. Those types of samples sizes might be perfectly reasonable for the histological measures but generally not for behavior. Moreover, the authors mentioned that rotarod data were collected longitudinally, but only average relative time on Rotarod from a single time point were shown. This makes it difficult for a reviewer to ascertain whether there is actually no behavioral influence. Similarly, relying on a single measure of motor coordination limits the ability to make claims. 

The burden of evidence for invalidating a model's reproducibility is larger and requires more statistical power than what exists here. Barring additional data I do not think this manuscript is suitable for publication. I highly recommend the authors further consider the notion that just as genetic etiologies for neurological diseases may not be fully penetrant, environmental toxicants only increase the risk for pathology. Accordingly, one would not expect all treated mice to go on the develop PD-like behavioral phenotypes. Acknowledgement of this is important and further exemplifies why a sample size of 3 (to 5) is uninterpretable. 

Author Response

Thank you for evaluating our manuscript and giving us your insight. We agree that the sample size at later time points is low. Unfortunately, we have obtained approval for only small sample sizes for this "model-establishment-experiment" (Institutional Review Board for Animal Ethics) and were requested to closely follow the published protocol.

Hence, we started the experiment with 6 animals in the control group and 10 animals in the rotenone group which were all observed up to 28 d, i.e. the period used in the previous studies within which strong rotenone effects were observed. At d28, half of the animals were killed for histological analysis and the remaining animals were treated for further 26 days. According to the original publication/protocol there should be clear rotenone effects after 28d in all animals (more or less) in terms of survival, Rotarod running, increase of a-Syn-positive neurons as well as a reduction of TH-positive neurons in the SN. To assess the biological/histologic effects, we had to euthanize half of the mice at this time point, and observed the remaining mice up to d54.

We agree and it is a good point that we should probably not expect 100% "success" of disease manifestation even with a toxic model. The crucial point is that rotenone was not bioavailable (no measurable plasma concentrations) with oral administration. Hence, the mild gliosis and low-grade systemic toxicity observed in the present study are likely caused by intestinal toxicity, but do not phenocopy Parkinson's Disease as claimed in previous studies. Gliosis was not described in the original publication, but was indeed observed in studies using oral rotenone for gut toxicity/ microbiome disruption, but TH neurons are unlikely to disappear without direct contact with rotenone. Hence, the discrepancy/non-reproducibility does not depend on 3 vs 6 animals at 54d.

As suggested we have changed the Methods and the Discussion accordingly and provide a justification for the low animal number in the later treatment period. 

Furthermore, we have added a number of diagrams including time courses showing the running behaviour of individual mice for each test day. In addition, we have added former data of a past control group of mice (Suppl. Fig. 1) which had similar ages, similar observation periods, replicate runs per test day, and identical genetic background. These mice show a similar running behaviour compared to rotenone and vehicle treated mice.

Round 2

Reviewer 2 Report

The justification for the small sample size helps the reviewer contextualize the results. The additional acknowledgement is critical so I appreciate that the manuscript now includes it. There remain a couple of issues that I would like to see addressed before I think the paper would be suitable for publication. 

The argument that the low bioavailability of rotenone in circulation likely underlies the findings. At what time point was blood sampled following dosing? More detail would be helpful. It is also worth mentioning here that the LOD of the analysis of 5 nM does not preclude pharmacodynamic effects as rotenone at sub 5 nM exerts effects. 

The term “ancient” controls is strange. The term “historic” or “previous” or something else might be better. 

Author Response

Thank you for again handling our manuscript and providing important recommendations.

The time point of blood drawing, which was 2h after the last rotenone dose, was already indicated in the “Methods” section. We have highlighted the sentence in yellow. Furthermore, we added the time point to the “Results” to make it easier for the readers to recognize it.

Concerning the LLOQ of 5 nM: With our LC-MS/MS instrument, it is possible to also detect very low plasma concentrations, between 0 and 5 nM (2 ng/mL). However, the concentration of the lowest standard is 5 nM, so that concentrations below this standard are approximate concentrations. The Lower Limit of Detection, LOD, as shown by our sample with 0.964 ng/mL is below the LLOQ. Since all of the plasma samples except the mentioned one showed no signal at all, it has to be assumed that there was no circulating rotenone.

We have replaced the term “ancient” by “previous” and “former” control mice, respectively.